

# SARS-CoV-2 Delta (B.1.617.2) variant replicates and induces syncytia formation in human induced pluripotent stem cell-derived macrophages

Theeradej Thaweerattanasinp, Asawin Wanitchang, Janya Saenboonrueng, Kanjana Srisutthisamphan, Nanchaya Wanasen, Suttipun Sungsuwan, Anan Jongkaewwattana and Thanathom Chailangkarn

Virology and Cell Technology Research Team, National Center for Genetic Engineering and Biotechnology (BIOTEC), National Science and Technology Development Agency (NSTDA), Pathum Thani, Thailand

## ABSTRACT

Alveolar macrophages are tissue-resident immune cells that protect epithelial cells in the alveoli from invasion by pathogens, including severe acute respiratory syndrome coronavirus 2 (SARS-CoV-2). Therefore, the interaction between macrophages and SARS-CoV-2 is inevitable. However, little is known about the role of macrophages in SARS-CoV-2 infection. Here, we generated macrophages from human induced pluripotent stem cells (hiPSCs) to investigate the susceptibility of hiPSC-derived macrophages (iMΦ) to the authentic SARS-CoV-2 Delta (B.1.617.2) and Omicron (B.1.1.529) variants as well as their gene expression profiles of proinflammatory cytokines during infection. With undetectable angiotensin-converting enzyme 2 (ACE2) mRNA and protein expression, iMΦ were susceptible to productive infection with the Delta variant, whereas infection of iMΦ with the Omicron variant was abortive. Interestingly, Delta induced cell-cell fusion or syncytia formation in iMΦ, which was not observed in Omicron-infected cells. However, iMΦ expressed moderate levels of proinflammatory cytokine genes in response to SARS-CoV-2 infection, in contrast to strong upregulation of these cytokine genes in response to polarization by lipopolysaccharide (LPS) and interferon-gamma (IFN-$\gamma$). Overall, our findings indicate that the SARS-CoV-2 Delta variant can replicate and cause syncytia formation in macrophages, suggesting that the Delta variant can enter cells with undetectable ACE2 levels and exhibit greater fusogenicity.

Corresponding author
Thanathom Chailangkarn,
thanathom.cha@biotec.or.th

## INTRODUCTION

The global pandemic of COVID-19 has fostered the mutation of SARS-CoV-2 into a number of variants of concern (VOCs). Despite widespread vaccination, the Delta (B.1.617.2) and Omicron (B.1.1.529) VOCs have caused a massive increase in COVID-19 cases and deaths worldwide. Several mutations in the spike proteins of the two VOCs, particularly L452R and P681R in Delta and E484A and Q498R in Omicron variants,

have been associated with enhanced viral transmission and high resistance to antibody neutralization (*Saito et al., 2022*; *Sharma et al., 2022*). However, the susceptibility of macrophages to these recently emerged VOCs, which are more infectious and could potentially infect cells even more efficiently than the original variant, remains unknown.

SARS-CoV-2 uses the spike protein to bind to ACE2 receptor on the cell surface to undergo cell entry (*Hoffmann et al., 2020*; *Shang et al., 2020*). Macrophages have been reported to express ACE2 on the cell surface (*Keidar et al., 2005*; *Song et al., 2020*), suggesting that SARS-CoV-2 could infect and replicate in macrophages *via* an ACE2-dependent pathway. However, other studies using single-cell RNA sequencing reported undetectable expression of ACE2 in macrophages (*Singh, Bansal & Feschotte, 2020*; *Liu et al., 2020*). Alternatively, infection of macrophages with SARS-CoV-2 could occur by phagocytosis of virions, followed by escape of the virus from the endolysosomal system *via* an ACE2-independent pathway (*Lv et al., 2021*). In addition, some studies suggest that alveolar macrophages can support the replication and release of SARS-CoV-2 (*Grant et al., 2021*; *Lv et al., 2021*), while others have demonstrated the abortive SARS-CoV-2 infection of macrophages (*Niles et al., 2021*; *Thorne et al., 2021*; *Zhang et al., 2022*). Therefore, it is unknown whether SARS-CoV-2 can replicate in macrophages.

In the lung, SARS-CoV-2 primarily targets and replicates in the alveolar epithelial type II cells lining the alveoli, where macrophages constitute the vast majority of resident immune cells (*Lamers & Haagmans, 2022*). As innate immune cells, macrophages engulf viral particles and release cytokines to control viral infection. However, whether SARS-CoV-2 could infect macrophages and hijack their cellular machinery as a reservoir to spread its viral progenies to extrapulmonary tissues and organs remains poorly understood. Only a few approaches are available to investigate macrophage susceptibility to SARS-CoV-2 infection, making it difficult to fully understand SARS-CoV-2 pathogenesis.

Derivation of macrophages from hiPSCs provides a new opportunity to study the role of macrophages in SARS-CoV-2 pathogenesis. Unlike human primary macrophages, which are typically obtained in non-uniform batches with limited cell number due to donor variability, iMΦ can be generated in large uniform batches with stable genotype and function (*Wilgenburg et al., 2013*). Rather than tissue-resident macrophages, iMΦ more closely resemble blood monocyte-derived macrophages (*Wilgenburg et al., 2013*), which are recruited to the lung during infection. We hypothesized that iMΦ could also become susceptible to infection with two distinct SARS-CoV-2 VOCs: Delta and Omicron, possibly eliciting gene expression of proinflammatory cytokines. Here, we generated and characterized iMΦ to investigate the susceptibility of iMΦ to Delta and Omicron variants as well as their gene expression profiles of proinflammatory cytokines during infection. In contrast to Omicron, Delta variant replicates and causes syncytia formation in iMΦ, suggesting that Delta variant might be able to enter host cells in an ACE2-independent manner and exhibit increased fusogenicity (*Saito et al., 2022*).

## MATERIALS & METHODS

### Cell cultures

Human induced pluripotent stem cells (hiPSCs; ATCC ACS-1019) were seeded in 6-well plates pre-coated with Matrigel (Cat# 354234; Corning, Corning, NY, USA). Cells were cultured in mTeSR-1 medium (Cat# 85850; StemCell Technologies, Vancouver, CA), which was changed daily. At about 70–80% confluency, cells were detached with Accutase (Cat# 07922; StemCell Technologies, Vancouver, CA) and passaged for maintenance or macrophage differentiation.

Human epithelial lung carcinoma cells stably overexpressing human ACE2 (A549-ACE2; a549d-cov2r; InvivoGen, San Diego, CA, USA), human hepatocellular carcinoma cells (Huh-7; ATCC PTA-4583), and human embryonic kidney epithelial cells (293T/17; ATCC CRL-11268) stably overexpressing human ACE2 (293T/17-ACE2) were cultured in DMEM (Cat# SH30243.02; Cytiva, Marlborough, MA, USA) with 10% fetal bovine serum (FBS; Cat# F7524; Sigma, St. Louis, MO, USA). Monkey kidney epithelial cells (VeroE6; ATCC CRL-1586) were cultured in Opti-MEM (Cat# 22600-050; Invitrogen, Waltham, MA, USA) with 10% FBS.

### Generation of hiPSC-derived macrophages

The protocol for generating hiPSC-derived macrophages (iMΦ) was adapted from previously published protocols (*Wilgenburg et al., 2013*; *Gutbier et al., 2020*), with minor modifications. On day 0, $4 \times 10^6$ cells of hiPSCs were initially stimulated to form embryoid bodies (EBs) by culturing them in a single well of an AggreWell 800 plate (Cat# 34850; StemCell Technologies, Vancouver, CA) with EB medium and incubating at 37 °C/5% $CO_2$. EB medium consisted of mTeSR-1 medium (Cat# 85850; StemCell Technologies, Vancouver, CA) supplemented with 10 μM Rock inhibitor (Y27632) (Cat# 72302; StemCell Technologies, Vancouver, CA), 50 ng/mL rhBMP4 (Cat# 120-05ET-10UG; PeproTech, Rocky Hil, NJ, USA), 20 ng/mL rhSCF (Cat# 78062.1; StemCell Technologies, Vancouver, CA), and 50 ng/mL rhVEGF (Cat# 100-20-10UG; PeproTech, Rocky Hill, NJ, USA). From days 1–3, the differentiated hiPSCs were replenished daily with the EB medium. On day 4, approximately 300 EBs were harvested into a 50-mL centrifuge tube containing differentiation medium. Differentiation medium consisted of X-VIVO 15 medium (Cat# BE02-053Q; Lonza, Basel, Switzerland); 100 ng/mL rhM-CSF (Cat# 216-MC; R&D Systems, Minneapolis, MN, USA); 25 ng/mL rhIL-3 (Cat# 203-IL; R&D Systems, Minneapolis, MN, USA); 2 mM Glutamax (Cat# 35050061; Gibco, Waltham, MA, USA); 1% Antibiotic-Antimycotic (Cat# 15240062; Gibco, Waltham, MA, USA); and 0.055 mM 2-mercaptoethanol (Cat# 21985023; Gibco, Waltham, MA, USA). Approximately 150 EBs could be cultured in a T175 flask pre-coated with Matrigel (Cat# 354234; Corning, Corning, NY, USA) using at least 15 mL of differentiation medium. Then, the cultured EBs were incubated at 37 °C/5% $CO_2$ without disturbing the flasks during the first week of differentiation. From week 2–3, at least 10 mL of fresh differentiation medium was added to each flask once per week. From week 4, half of differentiation medium for each flask was replaced with fresh differentiation medium once per week until macrophage precursors

appeared in the supernatant. Then, the entire differentiation medium was changed twice per week.

The floating macrophage progenitors were harvested and resuspended in macrophage medium. Macrophage medium consisted of X-VIVO 15 medium; 100 ng/mL rhM-CSF; 2 mM Glutamax; and 1% antibiotic-antimycotic. Macrophage precursors were cultured in macrophage medium at 37 °C/5% $CO_2$ for 7 days to obtain mature macrophages (iM$\Phi$) ready for use. For M1 (iM1$\Phi$) polarization, iM$\Phi$ were cultured for 24 h in X-VIVO 15 medium supplemented with 100 ng/mL LPS (Cat# L4391-1MG; Sigma, St. Louis, MO, USA); 20 ng/mL rhIFN $\gamma$ (Cat# 285-IF, R&D Systems; Minneapolis, MN, USA); 2 mM Glutamax; and 1% Antibiotic-Antimycotic. For M2 (iM2$\Phi$) polarization, macrophages were cultured for 24 h in X-VIVO 15 medium supplemented with 50 ng/mL rhIL-4 (Cat# 204-IL; R&D Systems, Minneapolis, MN, USA).

## Viruses

SARS-CoV-2 isolates were obtained from saliva samples of COVID-19 patients in Thailand and verified by nucleotide sequencing as the Delta (B.1.617.2; GISAID accession number EPI_ISL_11327790) and Omicron (B.1.1.529; EPI_ISL_11327789) variants. All experiments with SARS-CoV-2 were performed in a biosafety level 3 (BSL-3) laboratory at the National Center for Genetic Engineering and Biotechnology (BIOTEC), Thailand. Viruses were propagated in A549-ACE2 cells at 37 °C. Viral titers were measured using the median tissue culture infectious dose ($TCID_{50}$) assay on A549-ACE2 cells. For the $TCID_{50}$ assay, virus samples were serially diluted ten-fold (from 1:10 to $1:10^6$) in Opti-MEM (Cat# 22600-050; Invitrogen, Waltham, MA, USA) with 0.1% TrypLE Select (Cat# 0040090DG; Gibco, Waltham, MA, USA) and then incubated with confluent A549-ACE2 cells in 96-well plates for 72 h at 37 °C/5% $CO_2$. After incubation, cells were examined for cytopathic effects (CPEs) and confirmed by dot blots on nitrocellulose membranes (Bio-Rad, Hercules, CA, USA) with the following antibodies: primary anti-SARS-CoV-2 nucleocapsid (N) antibody (1:1000 dilution; in-house) and secondary goat anti-mouse IgG conjugated with HRP (1:4000 dilution; Cat# 405306; BioLegend, San Diego, CA, USA). Viral titers were calculated by the Spearman-Karber method and expressed as $TCID_{50}$/mL.

## Infection of viruses

iM$\Phi$ were infected with SARS-CoV-2 Delta or Omicron variant at a multiplicity of infection (MOI) of 0.1 in RPMI-1640 (Cat# 31800022; Gibco, Waltham, MA, USA) without serum. Mock-infected cells were used as controls. Cells were adsorbed with virus for 2 h, washed with PBS, replenished with RPMI-1640 supplemented with 0.05% TrypLE Select, and incubated at 37 °C/5% $CO_2$. Cell supernatants were harvested at 0, 24, 48 and 72 h post-infection (hpi) for virus titration. Cells were subjected to immunofluorescence assay to determine the expression of SARS-CoV-2 N protein.

## Immunofluorescence assay of viral proteins

Cells were fixed and permeabilized with 80% acetone for 15 min and blocked with 1% bovine serum albumin (BSA; Cat# A8412; Sigma, St. Louis, MO, USA) in PBST (PBS + 0.1%Tween 20) for at least 30 min. For SARS-CoV-2 detection, cells were stained with

primary mouse anti-SARS-CoV-2 N antibodies (1:1000 dilution; in-house) overnight at 4 °C and washed three times with PBS. Cells were then stained with secondary anti-mouse IgG antibodies conjugated with Alexa Fluor 568 (1:1000 dilution; Cat# A11004; Invitrogen, Waltham, MA, USA) and phalloidin conjugated with Alexa Fluor 488 (1:400 dilution; Cat# A12379; Invitrogen, Waltham, MA, USA) in the dark for at least 1 h at room temperature and washed three times with PBS.

Cell nuclei were counterstained with DAPI (Cat# H-1200; Vector Laboratories, Newark, CA, USA). Cells were imaged with an IX73 fluorescence microscope (Olympus, Tokyo, Japan) and analyzed with cellSens Standard Imaging Software (version 2.1; Olympus, Tokyo, Japan) and ImageJ (version 1.53c; Bethesda, MD).

## Flow cytometry

For surface marker analysis, macrophages were harvested with enzyme-free cell dissociation buffer (Cat# 13151014; Gibco, Waltham, MA, USA). Cells were resuspended in ice-cold FACS buffer (consisting of PBS and 0.5–1% BSA) at a concentration of $1\text{-}5 \times 10^6$ cells/mL. To block non-specific binding of the primary antibodies, Fc receptor blocking antibody (Cat# 422301; BioLegend, San Diego, CA, USA) was added to the cell suspension. Cells were incubated with the conjugated primary antibodies or the isotype-matched control in the dark for at least 30 min at 4 °C. Cells were washed three times with ice-cold FACS buffer before analyzing with a FACSAria Fusion flow cytometer (Becton Dickinson, Franklin Lakes, NJ, USA). Data were analyzed using FlowJo (version 10, Ashland, OR).

The following antibodies (all purchased from Invitrogen, Waltham, MA, USA) were used for staining surface markers: CD11b-APC (1:20 dilution; Cat# 17-0118-41), CD14-PE (1:20 dilution; Cat# 12-0149-42), CD16-FITC (1:10 dilution; Cat# 11-0168-41), CD86-PE (1:20 dilution; Cat# 12-0869-41), CD163-APC (1:20 dilution; Cat# 17-1639-41) and isotype controls: mouse-IgG1 kappa-APC (1:40 dilution; Cat# 17-47148-1), mouse-IgG1 kappa-PE (1:40 dilution; Cat# 12-4714-81), mouse-IgG1 kappa-FITC (1:200 dilution; Cat# 11-4714-81), mouse-IgG2b kappa-PE (1:40 dilution; Cat# 12-4732-81).

## Phagocytosis assay

Macrophage progenitors were cultured for 7 days in macrophage medium in a 96-well plate at a density of $1 \times 10^5$ cells/well. Mature iMΦ were incubated with or without reconstituted pHrodo Green Zymosan Bioparticles (Cat# P35365; Invitrogen, Waltham, MA, USA) at 37 °C for 2 h. Cells were harvested and analyzed by flow cytometry. Zymosan-free cells were used as negative controls to set a threshold for measuring the percentage of positive cells (*Wilgenburg et al., 2013*).

## Real-time quantitative reverse transcription PCR (RT-qPCR)

For human gene expression, total RNA was extracted from cell lysates using the GeneJET RNA Purification Kit (Cat# K0732; Thermo Fisher, Waltham, MA, USA). RT-qPCR was performed using the iTaq Universal SYBR Green One-Step Kit (Cat# 10032048; Bio-Rad, Hercules, CA, USA) on a CFX Opus 96 Real-Time PCR System (Bio-Rad, Hercules, CA, USA). Primers for ACE2, CD86, IL-1$\beta$, IL-6, IL-8, IL-18, TNF-$\alpha$, CCL2, IFN-$\alpha$, and GAPDH are described in Table S1. Cycling conditions were as follows: 1 cycle of 50 °C for

20 min, 1 cycle of 95 °C for 5 min; 40 cycles of 95 °C for 10 s and 58 °C for 30 s. Fluorescence signals were detected at the end of each 58 °C step. The dissociation curve was as follows: 1 cycle of 95 °C for 10 s, 65 °C for 5 s with a temperature increment of 0.5 °C to 95 °C. Ct values and dissociation curves were analyzed using CFX Maestro Software (Bio-Rad; Hercules, CA, USA, version 2.0). Gene expressions were normalized against GAPDH and compared with 293T/17 cells for the ACE2 gene or mock infection for cytokine genes using the $2^{-\Delta\Delta CT}$ method (*Schmittgen & Livak, 2008*). With the exception of ACE2 and CD86 gene expression, all other samples were measured in triplicate.

## Statistical analysis

All statistical tests were performed with GraphPad Prism 9 (San Diego, CA). One-way ANOVA with Tukey's post-hoc tests were used for multiple comparisons between experimental groups. Data are presented as mean ± SD. The *P* value < 0.05 was considered statistically significant.

## RESULTS

### Characterization of hiPSC-derived macrophages (iMΦ)

We generated iMΦ using previously published protocols (Figs. 1A–1B) (*Wilgenburg et al., 2013*; *Gutbier et al., 2020*). Differentiated iMΦ displayed round, oval, and spindle shapes (Fig. 1B), which are typical of macrophage morphology (*Rey-Giraud, Hafner & Ries, 2012*). Flow cytometry analysis of cell surface marker expression revealed that iMΦ expressed the monocyte/macrophage lineage markers CD11b, CD14, CD16, CD86 and CD163 (Fig. 2A). To assess phagocytic activity, iMΦ were incubated with pH-sensitive zymosan particles that fluoresce at an acidic pH typically found in phagosomes. After 2 h of zymosan incubation, the majority (>90%) of iMΦ engulfed the particles (Figs. 2B–2C), confirming that the cells were functional with their phagocytic activity. Taken together, the iMΦ generated in this study exhibit key characteristics of macrophages in terms of their morphology, surface marker expression, and phagocytic activity.

### iMΦ are susceptible to SARS-CoV-2 infection

Given that SARS-CoV-2 can infect mouse and human macrophages *via* ACE2-independent pathways (*Lv et al., 2021*; *Jalloh et al., 2022*), we first sought to investigate the susceptibility of iMΦ to SARS-CoV-2 infection. Under bright-field microscopy, we infected iMΦ with SARS-CoV-2 (Delta or Omicron variants) at an MOI of 0.1 and found no apparent signs of CPE at 72 hpi. In contrast, an immunofluorescence experiment revealed the presence of SARS-CoV-2 N protein in the cytoplasm of iMΦ infected with both Delta and Omicron (Fig. 3A; Fig. S1). SARS-CoV-2 N protein was detectable in 11.5% of Delta-infected iMΦ but only in 3.6% of cells infected with Omicron (Fig. 3B). Of note, we observed clear signs of cell–cell fusion or syncytia formation only in Delta-infected iMΦ (Fig. 3A; Fig. S1). It is worth noting that, despite the signs of viral entry, RT-qPCR and immunoblot analysis revealed undetectable mRNA and protein expression of ACE2 in iMΦ (Figs. S2A– Figs. S2B; Table S2). Viral replication was further examined by titrating infectious virions in iMΦ supernatants collected up to 72 hpi on A549-ACE2 cells using the $TCID_{50}$ assay (Fig.

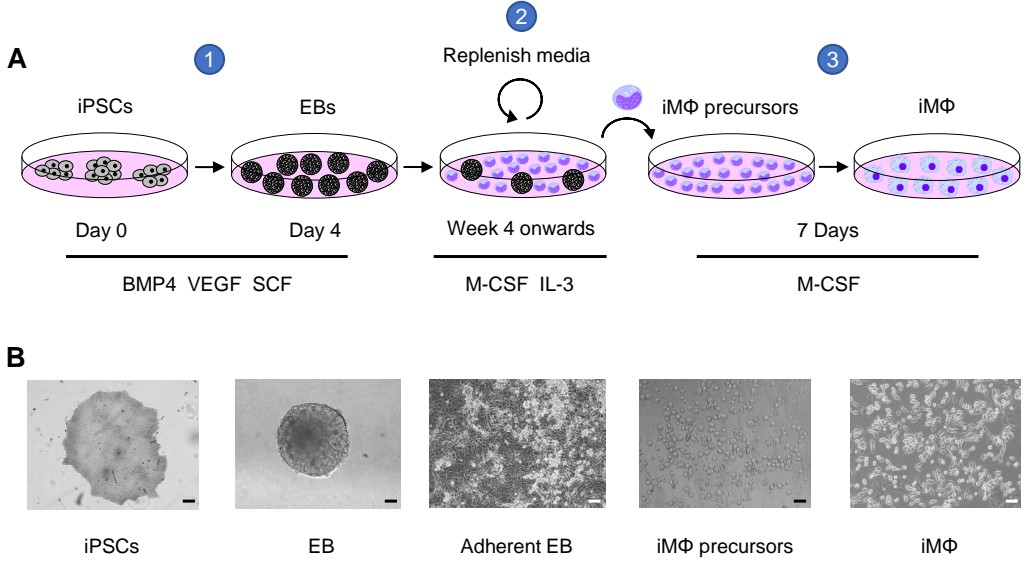

**Figure 1** **Macrophage differentiation protocol for hiPSCs.** (A) Schematic representation of human macrophage differentiation protocol. To induce EB formation, hiPSCs were cultured in the presence of BMP4, VEGF, and SCF (step 1). To produce iMΦ precursors, adherent EBs were cultured in the presence of M-CSF and IL-3 (step 2). The precursors differentiated into mature iMΦ in the presence of M-CSF (step 3). (B) Representative bright-field microscopy of human macrophage differentiation. Scale bars, 50 μm (EB; iMΦ precursors), 100 μm (iPSCs; adherent EB; iMΦ).

S3; Table S3). Replication of the Delta variant in iMΦ was productive within the first 48 hpi (Fig. 3C). Replication kinetics gradually increased from $8.73 \times 10$ TCID$_{50}$/mL at 0 hpi to $3.58 \times 10^2$ and $8.16 \times 10^2$ TCID$_{50}$/mL at 24 and 48 hpi, respectively (Fig. 3C). Later, however, viral titers dropped to $3.68 \times 10$ TCID$_{50}$/mL at 72 hpi (Fig. 3C). In contrast, Omicron replication in iMΦ was abortive, as we found no viral titer of the Omicron variant at any time point (Fig. 3C). Taken together, these results suggest that SARS-CoV-2 Delta variant replicates in iMΦ in an ACE2-independent manner and induces syncytia formation.

## iMΦ show moderate expression of proinflammatory cytokine genes in response to SARS-CoV-2 infection

We further investigated whether SARS-CoV-2 infection could activate iMΦ to overexpress proinflammatory cytokine genes, leading to overproduction of proinflammatory cytokines or cytokine storms as observed in severe COVID-19 patients (*Blanco-Melo et al., 2020*; *Hadjadj et al., 2020*; *Grant et al., 2021*). To this end, we examined mRNA expression of proinflammatory and antiviral cytokines in iMΦ infected with SARS-CoV-2 (Delta or Omicron) at 24, 48, and 72 hpi as well as in iMΦ polarized to the proinflammatory M1 phenotype by LPS and IFN-$\gamma$. After normalization to GAPDH using the $2^{-\Delta\Delta CT}$ method (*Schmittgen & Livak, 2008*), the results are expressed as a fold change in cytokine gene expression compared to mock infection (Fig. 4; Table S4). Although the infection rates of SARS-CoV-2 in iMΦ were approximately 11% and 4% for Delta and Omicron,

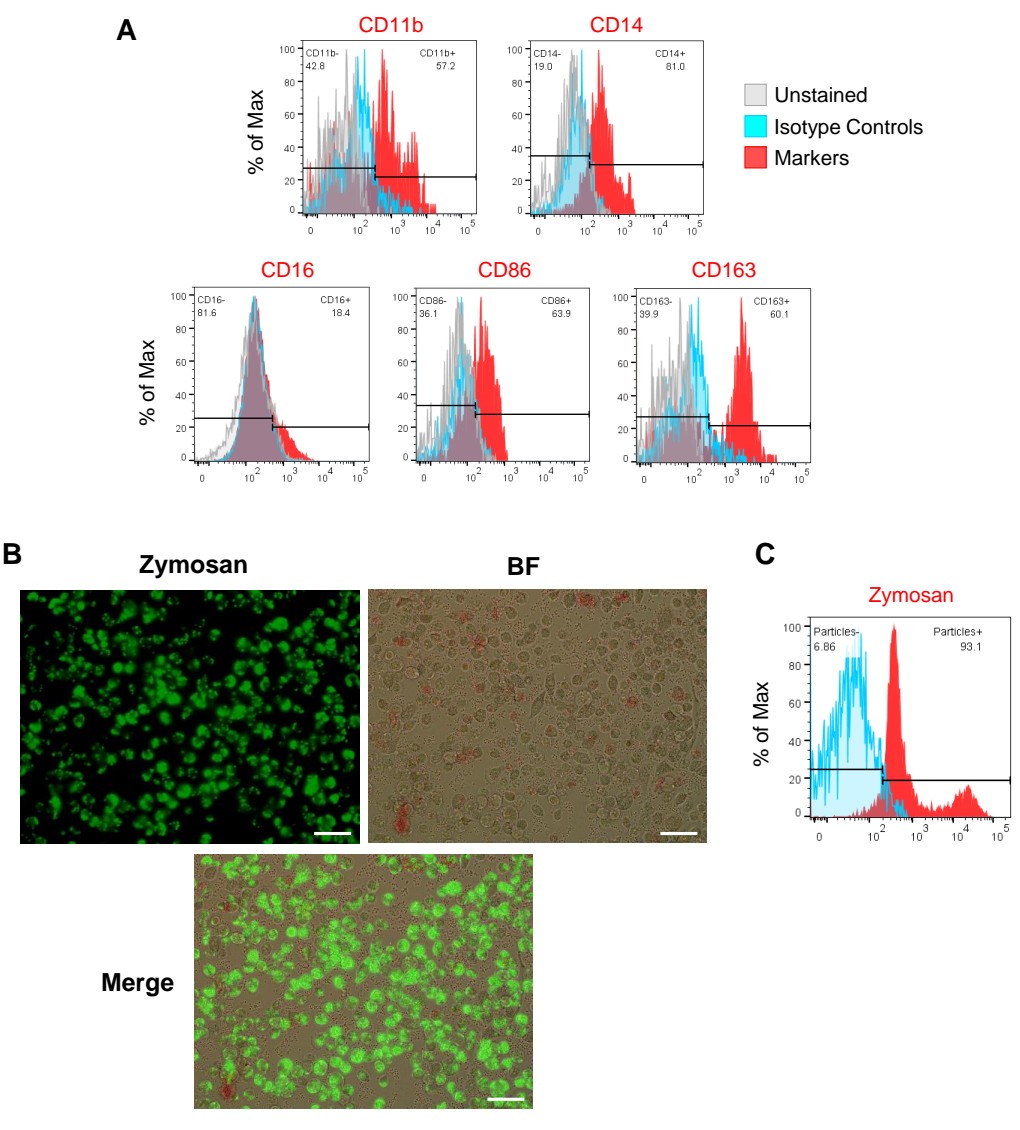

**Figure 2 Characterization of iMΦ.** (A) Cell surface marker expression of monocytic/macrophage cell lineage markers CD11b, CD14, CD16, CD86 and CD163 (red) analyzed by flow cytometry relative to isotype (blue) and unstained (grey) controls. (B) Composite fluorescence and bright-field microscopy of iMΦ phagocytosing pHrodo-Green zymosan particles (concentration: 100 μg/mL) at 2 h post-incubation. Scale bars, 50 μm. (C) Flow cytometry analysis of iMΦ (red) at 2 h post-incubation with pHrodo-Green zymosan particles relative to non-incubated cells (blue).

respectively (Fig. 3B), the infection did activate iMΦ to upregulate mRNA expression of the M1 phenotype marker CD86 (Fig. S4; Table S5).

Compared to mock controls, SARS-CoV-2 infection of iMΦ did not substantially upregulate cytokine mRNA expression (Figs. 4A–4G). Delta and Omicron infection only marginally altered mRNA expression of IL-1$\beta$ (Delta *vs* Mock: $p = 0.6114$; Omicron *vs* Mock: $p = 0.7534$), IL-6 ($p = 0.1910$; $p = 0.2619$), IL-18 ($p = 0.7755$; $p = 0.6092$), TNF-$\alpha$ ($p = 0.9080$; $p = 0.9405$) and IFN-$\alpha$ ($p = 0.0612$; $p = 0.2138$) (Figs. 4A–4B, 4D–4E, 4G).

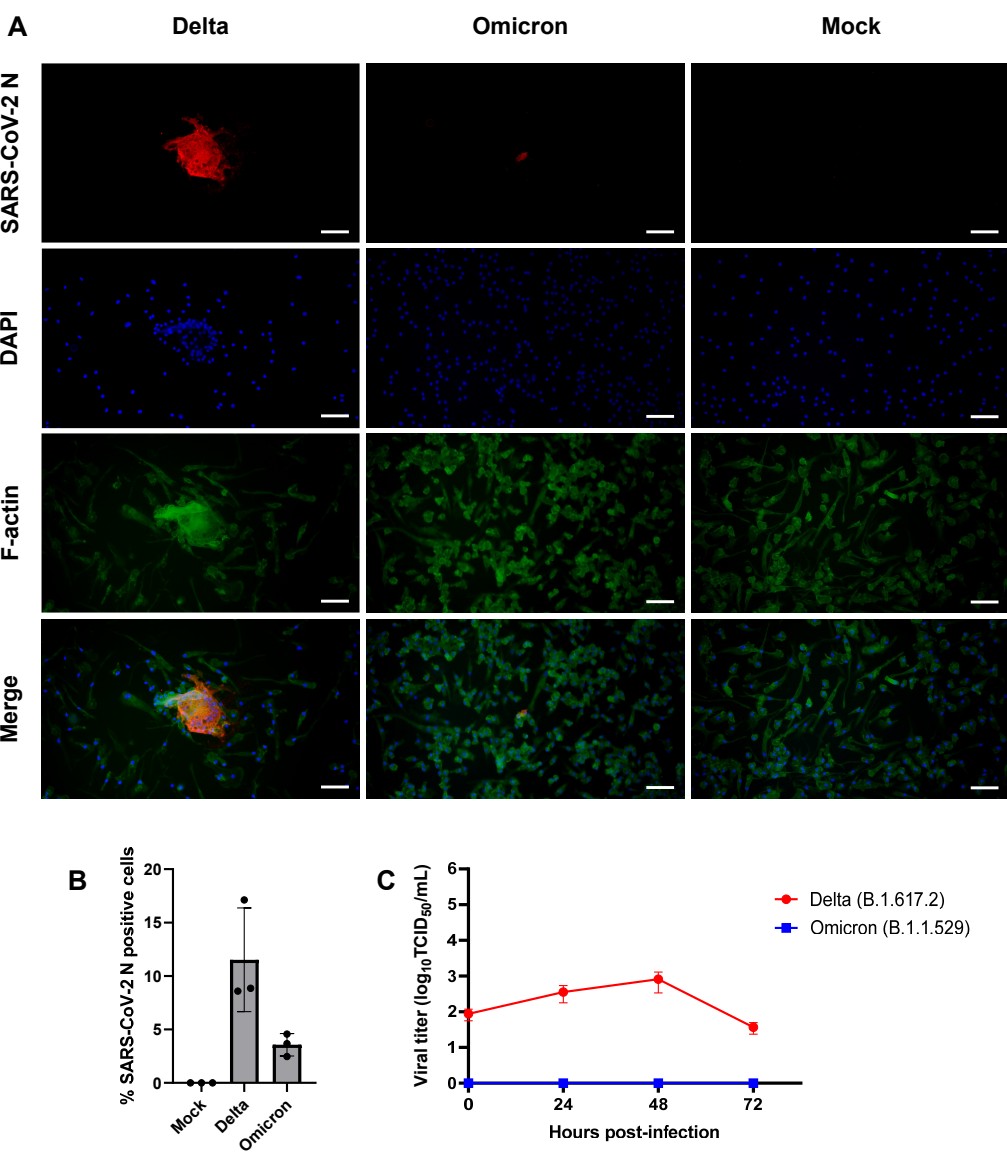

**Figure 3  SARS-CoV-2 Delta variant replicates and induces syncytia formation in iMΦ.** (A) Composite fluorescence microscopy of iMΦ infected with SARS-CoV-2 Delta and Omicron variants (MOI = 0.1). Cells were adsorbed with virus for 2 h and fixed at 72 hpi. Cells were stained with primary anti-SARS-CoV-2 N antibody followed by secondary Alexa Fluor 568-conjugated anti-mouse IgG antibody and Alexa Fluor 488-conjugated phalloidin. Cell nuclei were counterstained with DAPI. Scale bars, 50 μm. (B) Percentage of iMΦ positive for SARS-CoV-2 N after infection with SARS-CoV-2 Delta and Omicron variants. Data were shown as means ± SD ($n$ = 3 fields of view in each group). (C) Replication kinetics of SARS-CoV-2 Delta and Omicron variants in iMΦ. $TCID_{50}$ assay on A549-ACE2 cells was used to titrate the virus in the cell supernatants of Delta- or Omicron-infected iMΦ (MOI =0.1) at 0, 24, 48, and 72 hpi. Data were shown as means ± SD ($n$ = 3 in each group).

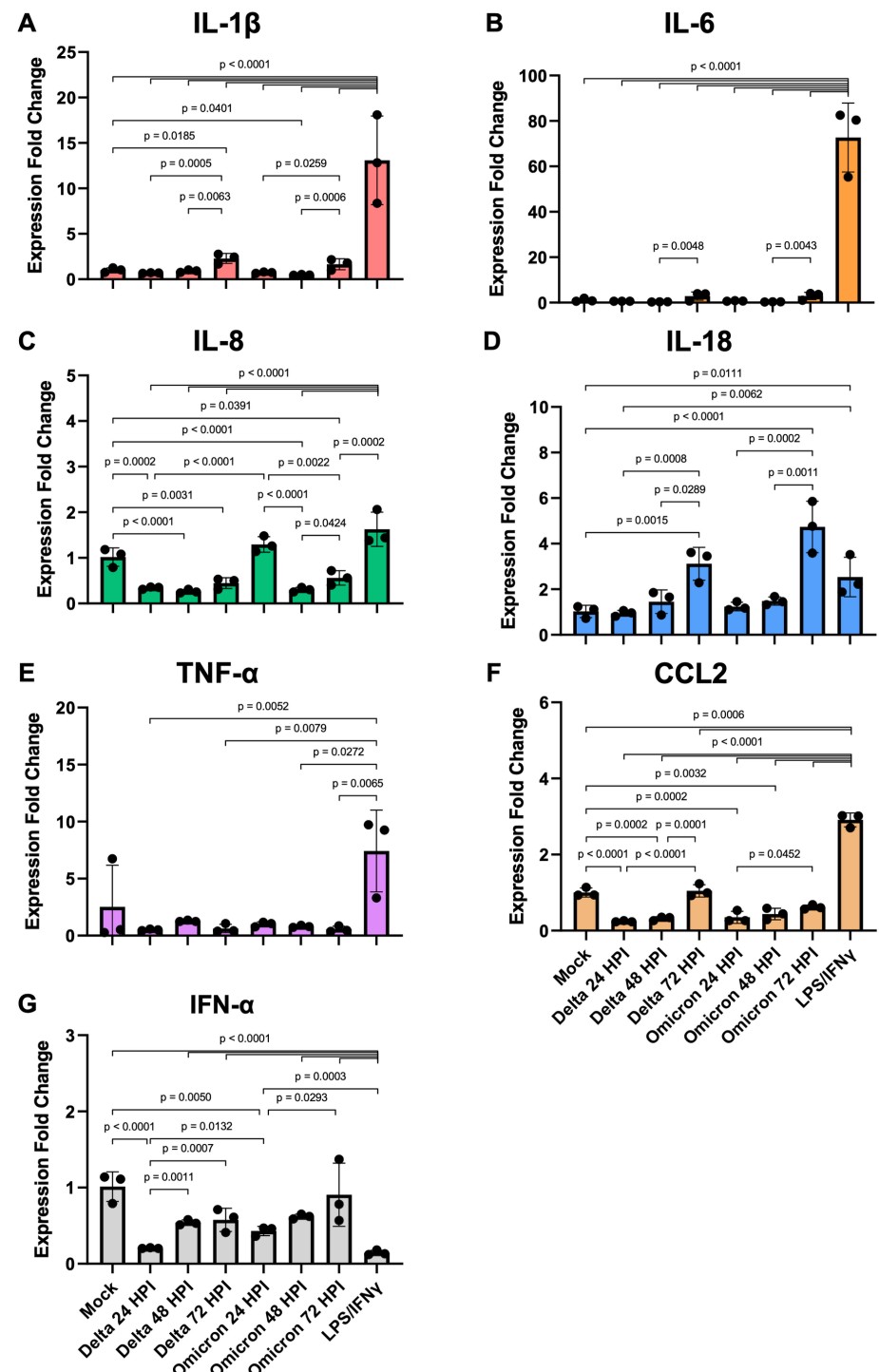

**Figure 4** **SARS-CoV-2 infection elicits moderate gene expression of proinflammatory cytokines from iMΦ.** Bar graphs display the relative gene expression fold change of (A) IL-1$\beta$, (B) IL-6, (C) IL-8, (D) IL-18, (E) TNF-$\alpha$, (F) CCL2, and (G) IFN-$\alpha$ mRNAs in iMΦ infected with SARS-CoV-2 Delta or Omicron variant (MOI = 0.1) and iMΦ pre-treated with lipopolysaccharides (continued on next page...)

**Figure 4 (…continued)**
(LPS) and interferon-gamma(IFN-$\gamma$). Total RNA was extracted from cell lysates at 24 h after LPS/IFN-$\gamma$ treatment or at 24, 48, and 72 hpi for SARS-CoV-2 infection. Gene expressions were quantified in triplicate by RT-qPCR. Data are expressed as fold change in cytokine gene expression compared with mock infection after normalization to GAPDH using $2^{-\Delta\Delta CT}$ method. Data were shown as means $\pm$ SD ($n = 3$ in each group). One-way ANOVA with Tukey's post hoc test on the log transformation of gene expression fold change.

Moreover, the infection suppressed mRNA expression of IL-8 ($\sim$3-fold; $p < 0.0001$; $p < 0.0001$) and CCL2 ($\sim$3-fold; $p < 0.0001$; $p = 0.0002$) (Figs. 4C, 4F).

However, among groups of SARS-CoV-2-infected iM$\Phi$, there was an overall trend toward a time-dependent increase in mRNA expression of most cytokines during infection. At 72 hpi, Delta infection upregulated at least two- to seven-fold mRNA expression of IL-1$\beta$ (Fig. 4A), IL-6 (Fig. 4B), IL-18 (Fig. 4D), CCL2 (Fig. 4F) and IFN-$\alpha$ (Fig. 4G), compared to those at 24 hpi (IL-1$\beta$: $p = 0.0005$; IL-18: $p = 0.0008$; CCL2: $p < 0.0001$; IFN-$\alpha$: $p = 0.0007$) and 48 hpi (IL-1$\beta$: $p = 0.0063$; IL-6: $p = 0.0048$; IL-18: $p = 0.0289$; CCL2: $p = 0.0001$). When infected with Omicron at 72 hpi, mRNA expression of the same group of cytokines also increased by similar magnitudes compared to those at 24 hpi (IL-1$\beta$: $p = 0.0259$; IL-18: $p = 0.0002$; CCL2: $p = 0.0452$; IFN-$\alpha$: $p = 0.0293$) and 48 hpi (IL-1$\beta$: $p = 0.0006$; IL-6: $p = 0.0043$; IL-18: $p = 0.0011$) (Figs. 4A–4B, 4D, 4F–4G). Interestingly, only at 24 hpi did Omicron infection induce significantly higher mRNA expression of IL-8 ($\sim$4-fold; $p < 0.0001$) and IFN-$\alpha$ ($\sim$2-fold; $p = 0.0132$) than Delta infection (Figs. 4C, 4G).

Compared to LPS/IFN-$\gamma$ polarization, SARS-CoV-2 infection resulted in significantly reduced mRNA expression of IL-1$\beta$ ($\sim$15-fold; Delta $vs$ LPS/IFN-$\gamma$: $p < 0.0001$; Omicron $vs$ LPS/IFN-$\gamma$: $p < 0.0001$), IL-6 ($\sim$108-fold; $p < 0.0001$; $p < 0.0001$), IL-8 ($\sim$4-fold; $p < 0.0001$; $p < 0.0001$), TNF-$\alpha$ ($\sim$10-fold; $p = 0.0052$; $p = 0.0272$) and CCL2 ($\sim$7-fold; $p < 0.0001$; $p < 0.0001$) (Figs. 4A–4C, 4E–4F). However, the infection generated significantly higher IFN-$\alpha$ mRNA expression ($\sim$4-fold; $p < 0.0001$; $p < 0.0001$) than non-viral polarization by bacterial LPS and type II IFN-$\gamma$ (Fig. 4G).

These results suggest that SARS-CoV-2 infection triggers moderate gene expression of proinflammatory cytokines in iM$\Phi$ at the transcriptional level. However, LPS/IFN-$\gamma$ polarization causes significant transcriptional activation of proinflammatory cytokine genes in iM$\Phi$. Thus, it is unlikely that a direct interaction between SARS-CoV-2 and macrophages is the main cause of increased proinflammatory cytokine production or cytokine storm.

# DISCUSSION

In the alveoli, tissue-resident alveolar macrophages serve as sentinels that provide immune surveillance and regulate tissue homeostasis (*Kosyreva et al., 2021*). Because SARS-CoV-2 can be transmitted *via* the respiratory tract, tissue-derived and recruited alveolar macrophages are thought to be the first immune cells to encounter the virus in the alveoli. However, isolating human alveolar macrophages for an *in vitro* study is a difficult task that can only be performed by trained clinicians performing bronchoscopy on sedated humans (*Collins et al., 2014*). Alternatively, monocyte-derived macrophages from blood samples

could be used to investigate the mechanisms of SARS-CoV-2 pathogenesis. However, due to donor heterogeneity, both sources of human primary macrophages provide relatively limited numbers of cells from each donor and inconsistent batches of macrophages. In this study, we successfully generated hiPSC-derived macrophages (iMΦ) that closely resemble recruited macrophages from blood monocytes (*Wilgenburg et al., 2013*), to investigate the susceptibility of iMΦ to SARS-CoV-2 Delta and Omicron variants as well as their gene expression profiles of proinflammatory cytokines during infection. Differentiated iMΦ displayed the typical macrophage morphology (Fig. 1) (*Rey-Giraud, Hafner & Ries, 2012*), expression of macrophage-specific markers (Fig. 2A) and phagocytic activity (Figs. 2B–2C), confirming the phenotypic and functional macrophage properties of iMΦ.

Although alveolar macrophages are thought to be the first to recognize and interact with SARS-CoV-2 in the alveoli, the role of macrophages in SARS-CoV-2 infection is still largely unknown. Previously, SARS-CoV-2 was shown to infect mouse alveolar macrophages *via* an ACE2-independent mechanism (*Lv et al., 2021*). Notably, SARS-CoV-2 was phagocytosed by M1-polarized macrophages, but the virus eventually escaped from the endosome to initiate viral replication in the cytoplasm (*Lv et al., 2021*). SARS-CoV-2 VOCs, including Delta and Omicron, are associated with increased transmissibility and infectivity (*Harvey et al., 2021*; *Saberiyan et al., 2022*). However, it is unclear how the increased characteristics of Delta and Omicron variants alter their interaction with macrophages. In the present study, we found SARS-CoV-2 N protein expression in 11.5% and 3.6% of iMΦ infected with Delta and Omicron variants, respectively (Figs. 3A–3B). However, only Delta infection of iMΦ resulted in the release of infectious virions during the first 48 h (Fig. 3C). This conclusion is inconsistent with previous studies showing that using at least an MOI of 0.1, infection of macrophages with SARS-CoV-2 is abortive (*Niles et al., 2021*; *Jalloh et al., 2022*; *Zhang et al., 2022*). However, since most of these studies used old SARS-CoV-2 isolates for viral infection, the results are likely to be different. Another point worth noting is that iMΦ do not express detectable levels of ACE2 (Fig. S2), making it unlikely that SARS-CoV-2 infection in this cell type is dependent on ACE2 receptor. Surprisingly, Delta infection of iMΦ induced the formation of syncytia in iMΦ (Fig. 3A; Fig. S1). This observation points to the possibility that syncytia formation may increase cell-to-cell transmission of the SARS-CoV-2 Delta variant in nearby macrophages, allowing the virus to avoid neutralizing antibodies in the extracellular space (*e.g.*, alveolar space) that may prevent cell-free infection (*Rajah et al., 2022*). Other respiratory viruses, such as measles, influenza, and respiratory syncytial virus, also utilize syncytia, formation for more efficient and rapid viral dissemination (*Cifuentes-Muñoz, Dutch & Cattaneo, 2018*; *Rajah et al., 2022*). However, because Omicron variant could infect significantly fewer cells than Delta (Fig. 3B), it is possible that replication and syncytia formation of Omicron in iMΦ would be less likely to be observed. Consequently, macrophages could serve as a viral reservoir for systemic dissemination of the SARS-CoV-2 Delta variant *via* syncytia formation. However, further research is needed to understand the molecular mechanisms and potential functions of syncytia formation in SARS-CoV-2-infected macrophages.

The immune responses of alveolar macrophages, which are constantly exposed to the outside atmosphere, must be tightly regulated to fight viral infections while minimizing
tissue damage and maintaining normal pulmonary function (*Divangahi, King & Pernet, 2015*). Alveolar macrophages are the major producers of type I IFNs in response to viral infections in the lung (*Kumagai et al., 2007*; *Divangahi, King & Pernet, 2015*). However, SARS-CoV-2 infection has been shown to suppress type I IFN antiviral responses (*e.g.*, IFN-$\alpha$ and IFN-$\beta$) and elevate the production of inflammatory cytokines (*e.g.*, IL-1$\beta$, IL-6, and TNF-$\alpha$) in blood and lung samples from COVID-19 patients (*Blanco-Melo et al., 2020*; *Hadjadj et al., 2020*; *Grant et al., 2021*). How SARS-CoV-2 infection affects alveolar macrophage cytokine responses remains largely unclear. In the present study, SARS-CoV-2 infection did not strongly upregulate mRNA expression of proinflammatory and antiviral cytokines in iM$\Phi$ (Fig. 4). However, SARS-CoV-2-infected iM$\Phi$ increasingly expressed most cytokine mRNAs over time from 24 to 72 hpi during infection (Figs. 4A–4B, 4D, 4F–4G). At 24 hpi, Omicron infection induced significantly higher IL-8 and IFN-$\alpha$ mRNA expression in iM$\Phi$ than Delta infection (Figs. 4C, 4G), which could in part contribute to the observed differences in viral replication and syncytia formation between the two variants. However, total proinflammatory cytokine gene expression in response to SARS-CoV-2 infection was at a significantly lower level than LPS/IFN-$\gamma$ polarization of iM$\Phi$ (Figs. 4A–4C, 4E–4F). Moreover, SARS-CoV-2 infection barely triggered antiviral type I IFN-$\alpha$ gene expression in iM$\Phi$ (Fig. 4G). These results are consistent with previous studies reporting moderate proinflammatory and antiviral cytokine responses following SARS-CoV-2 infection of primary macrophages (*Niles et al., 2021*; *Thorne et al., 2021*; *Zhang et al., 2022*). Therefore, it is unlikely that direct interaction between SARS-CoV-2 and macrophages is the main reason for the excessive production of proinflammatory cytokines or cytokine storm during early infection. Instead, secreted inflammatory mediators from infected lung epithelial cells may primarily drive macrophages to exacerbate inflammatory responses during the later stages of SARS-CoV-2 infection (*Thorne et al., 2021*; *Zhang et al., 2022*). However, without direct measurement of cytokine levels in the cell culture supernatants in this study, we cannot rule out the possibilities of robust proinflammatory cytokine release after SARS-CoV-2 infection of iM$\Phi$ and differential cytokine response profiles between Delta and Omicron infection.

## CONCLUSIONS

We generated hiPSC-derived macrophages and demonstrated that they are susceptible to productive infection with SARS-CoV-2 Delta variant, probably *via* an ACE2-independent pathway. The Delta variant also induces syncytia formation in these immune cells, supporting the enhanced fusogenicity of this SARS-CoV-2 variant. However, SARS-CoV-2 infection triggers only moderate gene expression of proinflammatory cytokines in macrophages. The findings suggest that other exogenous stimuli may be the main cause of excessive cytokine production in macrophages during the early phase of SARS-CoV-2 infection. Further *in vivo* studies are needed to elucidate the molecular mechanisms of the SARS-CoV-2-induced cytokine storm in the lung and systemic inflammation in multiple organs.

## ACKNOWLEDGEMENTS

We thank all the members of the Virology and Cell Technology Laboratory for their technical assistance and comments on this work.

### Funding

This work was supported by the National Center for Genetic Engineering and Biotechnology, Thailand (P-18-50193) to Thanathom Chailangkarn, and the National Science and Technology Development Agency, Thailand (P-22-51086) to Anan Jongkaewwattana. The funders had no role in study design, data collection and analysis, decision to publish, or preparation of the manuscript.

### Grant Disclosures

The following grant information was disclosed by the authors:
National Center for Genetic Engineering and Biotechnology, Thailand: P-18-50193.
National Science and Technology Development Agency, Thailand: P-22-51086.

### Competing Interests

The authors declare there are no competing interests.

### Author Contributions

- Theeradej Thaweerattanasinp conceived and designed the experiments, performed the experiments, analyzed the data, prepared figures and/or tables, authored or reviewed drafts of the article, and approved the final draft.
- Asawin Wanitchang performed the experiments, prepared figures and/or tables, and approved the final draft.
- Janya Saenboonrueng performed the experiments, prepared figures and/or tables, and approved the final draft.
- Kanjana Srisutthisamphan performed the experiments, prepared figures and/or tables, and approved the final draft.
- Nanchaya Wanasen performed the experiments, analyzed the data, authored or reviewed drafts of the article, and approved the final draft.
- Suttipun Sungsuwan performed the experiments, analyzed the data, authored or reviewed drafts of the article, and approved the final draft.
- Anan Jongkaewwattana conceived and designed the experiments, performed the experiments, authored or reviewed drafts of the article, and approved the final draft.
- Thanathom Chailangkarn conceived and designed the experiments, performed the experiments, authored or reviewed drafts of the article, and approved the final draft.

### Data Availability

 The raw data is available in the Supplementary Files.

## Supplemental Information

Supplemental information for this article can be found online at http://dx.doi.org/10.7717/peerj.14918#supplemental-information.

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
