# Peer review of "SARS-CoV-2 Delta (B.1.617.2) variant replicates and induces syncytia formation in human induced pluripotent stem cell-derived macrophages"

_PeerJ, doi:10.7717/peerj.14918_

## Round 0.1 · original submission · Major Revisions

Please pay attention to the review comments from the two reviewers, especially Reviewer 1's comments.

Reviewer 1 ·

Basic reporting

In this study, the authors suggested that iPSC-derived macrophages could be infected by SARS-CoV-2 delta variant virus independent of ACE2. This variant instead of omicron variant formed syncytia of macrophages and replicated. The manuscript of this study uses understandable and clear English.
Introduction of the study has described the background of the study and well cited the current research, however, I would suggest that the content of paragraph 3 (from line 64) should be at the beginning of the introduction, followed by content of paragraph 2 (from line 51) and paragraph 1 (from line 42), with the above structure, the introduction would be more clear and logical. The hypothesis of the authors is not clearly stated in the introduction thus it is not able to link the hypothesis and results of the study.
In this manuscript, figures are well shown and labeled. Authors provided figures included in the results however not all the results had confident figures (e.g. viral titer showed no figure of the staining, only statistical plot showed without triplicates). In addition, authors did not mention the availability of the raw data.
Discussion and conclusion of the manuscript covered the majority of the results, however, due to unclear research questions and hypotheses, it was still not solid to conclude the findings of the study.
This manuscript showed interesting results and conclusions, however, it seems that it is not a complete report that there are still some key points missing. Therefore, I would suggest a major revision of this manuscript.

Experimental design

This manuscript fits the aim and scope of PeerJ, the topic of this study is interesting and meaningful, however, the authors did not state clearly the research question: Is this study aim to reveal the mechanism of infectivity of SARS-CoV-2 of tissue-resident-macrophages? (line 48-50) or susceptibility of VOC than wt-virus (line 70-72)? However, these questions were not answered in the manuscript.
The study's design has a major missing piece: there is no comparison of iPSC-derived macrophage to primary-(tissue-resident or alveolar) macrophage when authors claimed that iPSC-derived macrophage is a good model to mimic the primary macrophage without batch variations. Another major defect of this study is replication of the experiments, all experiments in the study were only duplicates, this is UNSCIENTIFIC and UNACCEPTABLE. It is impossible to perform statistical analysis with duplicated results, even though the differences were small. Please increase to at least 3.
There are still some minor weak points of the study to be improved:
1. When M1/M2 was differentiated, please also check the marker of them. It is also interesting to compare infected macrophages and non-infected M1/M2 to identify the polarization of infected macrophages. Is there any sign of macrophage activation after infection? CD86 staining after infection can be included to show the activation status of macrophages.
2. What is the reason to perform the phagocytosis assay? Are there any changes in the phagocytosis ability before and after infection?
3. Why MOI here is 0.1? Are there any preliminary data to show this is the proper MOI?
4. Here in the study, qPCR was used to determine cytokine level, I would suggest that ELISA of cell culture supernatant should be used to have quantitative analysis.
5. Are there any reasons for qPCR using geometric mean rather than arithmetic means?

Validity of the findings

The findings of this study give a clue that the SARS-CoV-2 delta variant infects macrophages in an ACE2-independent manner. However, these results are not able to conclude as in the manuscript. Here, there are still questions and concerns for the authors to solve:
1. The immunocytochemistry showed only cells infected with Delta variant formed syncytia but not Omicron variant. How many fields/views have been counted? Since only DAPI and anti-N protein were used, is it possible to stain CD11b/CD14 or at least F-actin to visualize that cells were merged only with Delta infection.
2. Why qPCR was done at 72 hpi? It is shown in viral titer that after 48 hpi, viral titer was highest, please consider cytokine measurement at 48 hpi or earlier. It would even better if a time-course can be included to show the dynamic change.
3. Are there any reasons to use IFNβ and CXCL10 as biomarkers after infection? IFNβ is more secreted by fibroblast and very little in macrophages. For qPCR results of this study, IFNβ changes were really small. I would suggest using IFNα rather than IFNβ. For CXCL10 (10 kDa interferon-gamma-induced protein), it is induced by IFNγ, this has been clearly shown in the LPS/IFNγ control cells (~1000 fold), but in the infected cells, no IFNγ was used to prime cells, it can be expected that CXCL10 had no/low change in this case. For chemokine, I would suggest using IL-8/CXCL8 or CXCL1.
4. Cytokines/chemokines chosen in this study were not very macrophage-specific, IL-6 and TNF are nice as indicators for macrophage response, but it would be better to include IL-1β and IL-18 as macrophage-specific cytokines.
Considering the above questions and concerns, I would still doubt the conclusions from the authors when chosen biomarkers were not correct, but I am still very optimistic about this study if the authors improve and update their study to the manuscript.

Reviewer 2 ·

Basic reporting

The manuscript titled "SARS-CoV-2 Delta (B.1.617.2) variant replicates and induces syncytia formation in human induced pluripotent stem cell derived macrophages" is well written in a good structure containing all necessary parts. However, the English writing need to be improved to make the manuscript more clear and professional.

Experimental design

1. The author needs to consider the structure and paragraph order of introduction. For example, it is more clear to give a comprehensive overview of COVID-19 pandemic before introducing research findings on SARS-CoV-2 infection and role of alveolar macrophages.

2. The author described that there is great limitations on the isolation of primary human alveolar macrophages. In the study, author had successfully generated macrophages from human induced pluripotent stem cells(hiPSCs), which provided a good in vitro model to study the interaction between SARS-CoV-2 infection and macrophages.

3. The author highlighted the vital role of tissue resident alveolar macrophages in lung tissue on defensing viral infection. Thus, hiPSC-derived macrophages were generated and use to mimic alveolar macrophages. However, a clear comparison of them on expressing specific markers need to be described to valid the establishment of the in vitro model.

Validity of the findings

1. In the whole manuscript, the author used macrophages or alveolar macrophages but did not make it clear that whether you are focusing on tissue-resident alveolar macrophages or recruited macrophages? What your hiPSC-derived macrophages indeed stand for? What is your exact target macrophages? In the first result part, author showed that these cells are expressing monocyte/macrophage markers CD11b, CD14, CD16, CD86 and CD163. Is there any other evidences to show that they are alveolar macrophages? Markers like CD206, CD169 are recommended to investigate.
Furthermore, the author discuss alveolar macrophages again in discussion part. Thus, there is a misleading and misuse of macrophages and alveolar macrophages. Please: 1) make sure what is the target macrophages of the study; 2) prove whether hi-PSC-derived macrophages are the target.

2. In Figure 2C, zymosan was used to investigate the phagocytosis of hiPSC-derived macrophages and the flow cytometry analysis revealed that there were two peaks of particle uptake by macrophages, what are the differences in between?

3. In line 234, author need to explain why SARS-CoV-2 (Delta or Omicron variants) at an MOI of 0.1 was selected and used in the study.

4. Author showed that SARS-CoV-2 Delta variant but not Omicron variant induced syncytia formation in hiPSC-derived macrophages indicating the ability of virus replication of Delta. However, no significant size differences of SARS-CoV-2 N protein expression was shown in Figure 3A. Any quantification was done here to prove? Besides, the current IF staining was not enough to prove the syncytia formation. Markers of cell morphology is needed to include for staining.

5. As shown in Figure 3A, the peak of Delta virus titer was 48 hpi whereas there was a decrease of virus titer after 72 h. Why 72 h was selected as the time point to study the pro-inflammatory response induced by SARS-CoV-2 infection in hiPSC-derived macrophages? Early time points like 24 h or 48 h seems to be better than 72 h. For example, TNF-α is an early responsive gene, 72 h may be late to investigate the transcriptomic alteration.

6. Author used LPS and IFN-γ treatment as the macrophages M1 polarization positive control. Can SARS-CoV-2 infection induced M1 polarization in the current study? Author needs to explain and provides evidence on this issue.

7. Since author only studied the gene expression but not cytokine release into supernatant after SARS-CoV-2 infection, I recommend to use pro-inflammatory gene expression instead of pro-inflammatory cytokine responses in Figure 4.

8. How to explain the decrease of CCL2, CXCL10 and IFN-β? Again, time points are important to in the experiment setting to investigate the pro-inflammatory responses.

9. What do we conclude from the finding that, Omicron showed no ability to replicate and induce syncytia formation? This is necessary to be included in discussion.

Additional comments

1. In the introduction part, author described that SARS-CoV-2 infects in cells of the alveolar epithelium type II, please use official name ”alveolar epithelial type II cells”.

2. The selection of analysis time points seems to be a reason for the lack of pro-inflammatory responses evidence induced by SARS-CoV-2 infection.

3. In line 260, the statistical analysis method is not necessary to mention here.

4. Please clarify the focus of macrophage types in the study, author needs to check the whole manuscript carefully and make a clear structure.

---

## Round 0.2 · Minor Revisions

Please see to the two reviewers for their comments on your manuscript, specifically on Figures 3 and 4 as well as the data results they are representing.

Reviewer 1 ·

Basic reporting

The revision answer the questions from last review, I would be satisfied with the answers. I would give a minor revisions before accepting the manuscript.

Experimental design

No more comments about the expeimental design.

Validity of the findings

I am happy that the authors have updated new finding in this version, but I have some more questions that authors might answer in the manuscript before final version. Please in the next revison explain with a few sentence why in Fig3 it looks syncytia formated after omicron infection but no replication? if you think it is not a syncytica, please use another image rather than current one to avoid misunderstanding? I am also curious why cytokine response was so similar after delta and omicron infection but replication and syncytia formation were different, do you have more explaination on this?

Additional comments

Please remove the unnecessary comparisions in Fig.4 (e.g. delta 72hpi vs omicron 24hpi), just keep the comparison between same variants different time points or same time points different variants. For p value, please use same style, either accurate number (e.g. P=0.0132) or asterisk (e.g. **** P< 0.001 ).

Reviewer 2 ·

Basic reporting

Authors have taken all comments into consideration and modified through the whole manuscript. Besides, some more experiments were also conducted to prove findings.

Experimental design

no comment

Validity of the findings

There are two small issues should be explained. In Figure 3, immunofluorescence staining showed that Delta cause syncytia formation. Does Omicron have similar effect, even it is quite small according to the representative images? Any quantification was performed to measure the syncytia size? Please give a simple explanation.

In Figure 4, only necessary comparisons are needed to show significance, to make the plots easier to visualize. According to the qPCR result, no difference of proinflammatory gene expression patterns was found between Delta and Omicron. Can you explain why and how can it links to the syncytia formation?

Additional comments

no comment

---

## Round 0.3 · accepted · Accept

I believe that the authors have addressed all of the reviewers' comments.

Reviewer 1 ·

Basic reporting

I am satisfied with the revison of the manuscript, I would accept it.

Experimental design

no more comment

Validity of the findings

no more comment

Reviewer 2 ·

Basic reporting

Authors consider all the comments and have given clear explanations.

Experimental design

no comment

Validity of the findings

no comment

Additional comments

no comment